# Tactile-Driven Dexterous In-Hand Writing via Extrinsic Contact Sensing

Can Zhao[1], Lingzi Xie[2], Bidan Huang[2*], Shuai Wang[2], and Daolin Ma[1]

*Abstract*— Dexterous in-hand manipulation, especially involving interactions between grasped objects and external environments, remains a formidable challenge in robotics. This study tackles the complexities of in-hand manipulation under extrinsic contact through a representative three-finger handwriting task. We propose a hybrid arm-hand coordination framework that combines reinforcement learning with compliance control, offering both flexibility and robustness. Leveraging tactile sensors embedded in each finger, our tactile-driven estimation model dynamically predicts in-hand object pose and external contact, eliminating the need for fixed contact states. The proposed framework is first validated in simulation, where it successfully executes diverse writing tasks with accurate contact sensing. Sim-to-Real transfer is achieved through systematic calibration of finger joints and tactile sensors, supported by domain randomization. Real-world experiments further demonstrate the system's adaptability to writing tools with varying physical properties—such as radius, length, mass, and friction—while maintaining stability across different trajectories. Also see **https://inhandwriting. github.io/**. This work advances robotic manipulation capabilities in unstructured environments.

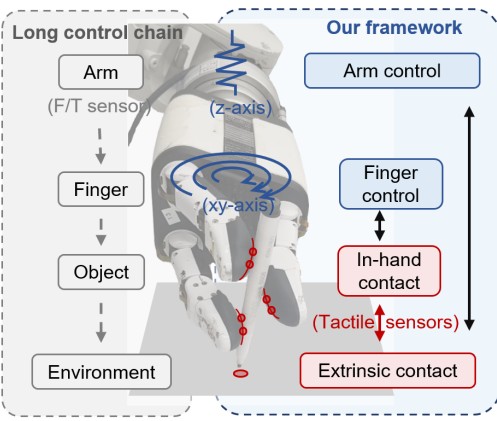

Fig. 1. Overview of our framework (right), which utilizes tactile-based feedback for direct and precise contact sensing, in contrast to the indirect wrist-mounted force/torque sensing (left).

## I. INTRODUCTION

Dexterous in-hand manipulation with extrinsic contact is vital for numerous real-world robotic applications, such as tool usage (e.g., fastening screws with a screwdriver) and component assembly (e.g., aligning and fitting gears in mechanical assemblies). These tasks demand simultaneously managing in-hand dynamics while maintaining stable contact between the manipulated object and the external environment. Although notable advancements have been made in learning-based approaches that utilize vision, proprioception, and tactile sensing for in-hand manipulation tasks [1]–[4], most state-of-the-art methods assume that the manipulated object remains isolated from extrinsic contact. However, interactions with the environment are often inevitable in practical scenarios, and these external contacts frequently play a critical role in task success [5], [6]. Recent studies have begun exploring extrinsic contact sensing and control, but the prevalent reliance on fixed gripper-object contact constrains the flexibility and practicality of these methods [7], [8].

A representative task of in-hand manipulation with extrinsic contact is **In-hand writing**, where a robotic hand dynamically controls a tool (e.g., a pen) while interacting with an external surface [9]. During writing, the robotic hand must coordinate multi-finger joints to maintain a stable grasp on the writing tool and follow reference trajectories (**In-hand contact control**), while adjusting the tool's contact with the writing surface (**Extrinsic contact control**). Successfully achieving this dual-level control introduces three major challenges: i) estimating and controlling both in-hand and extrinsic contact states simultaneously; ii) ensuring a secure grasp on the tool while dynamically adapting to variations in tool properties and external disturbances; and iii) coordinating multi-finger and wrist movements to balance stability and adaptability.

To address these challenges, we propose a tactile-driven framework that synergizes in-hand and extrinsic contact control through a hybrid arm-hand coordination strategy. Our approach leverages rich tactile feedback from fingertip sensors on a three-finger robotic hand to predict continuous object states and external contact conditions. As described in Fig. 1, fingertip tactile sensing, which directly interacts with the manipulated object, provides more localized and detailed contact dynamics than wrist-mounted force/torque sensors. This immediate feedback is essential for maintaining fine-grained contact control during dexterous in-hand manipulation tasks, improving both precision and robustness. By analyzing these tactile signals, we extract key features that enable reinforcement learning (RL) to effectively regulate in-hand contact and trajectory tracking. Our framework further integrates RL-based finger motion control with compliant position-force wrist adjustments, ensuring stable yet adaptive contact with external surfaces.

This work was supported by the National Natural Science Foundation of China (No. 12272220), and by the Tencent Robotics X. This work was done when C. Zhao and L. Xie were interns at Tencent Robotics X. *(Corresponding author: Bidan Huang).*

C. Zhao, D. Ma are with the School of Ocean & Civil Engineering, Shanghai Jiao Tong University, Shanghai 200240, China (e-mail: can.zhxx, daolinma@sjtu.edu.cn). L. Xie, B. Huang, and S. Wang are with Tencent Robotics X, Shenzhen 518054, China (e-mail: lingzixie, bidanhuang, shawnshwang@tencent.com).

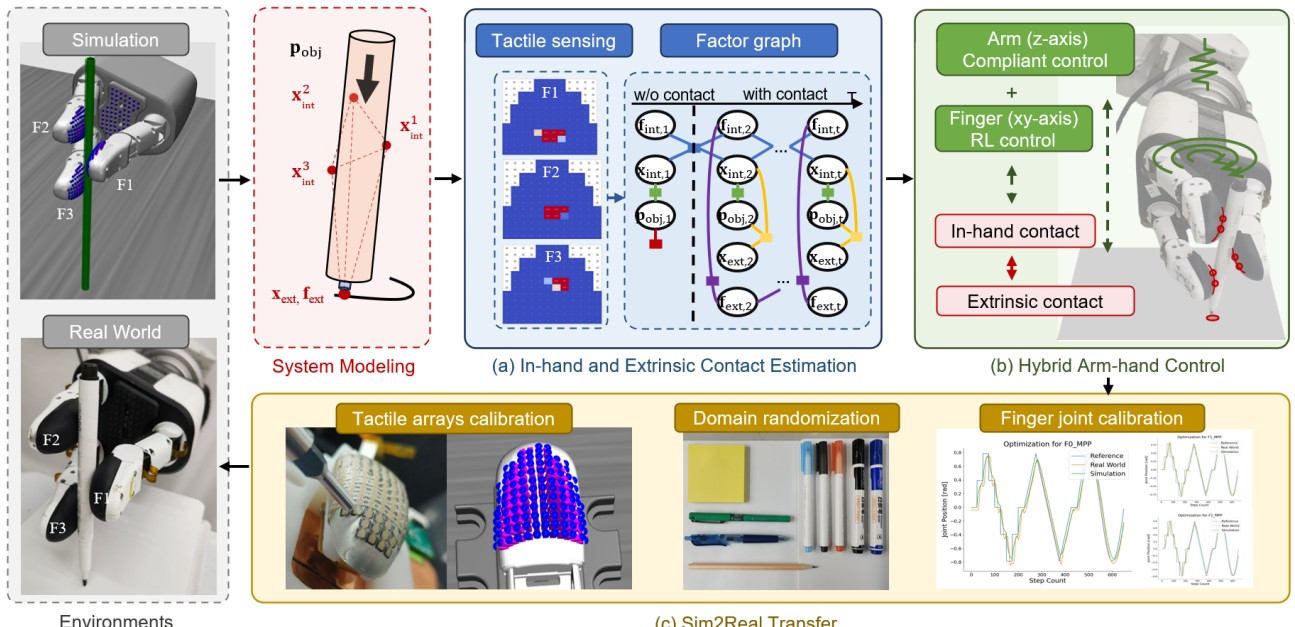

Fig. 2. Overall pipeline. (a) In-hand and extrinsic contact estimation via factor graph; (b) hybrid arm-hand coordination control framework that combines fingers' RL policy with wrist's compliant control; (c) sim-to-real transfer for real-world in-hand writing, including system identification and domain randomization.

## II. METHOD

Our method consists of three core components for tactile-based writing control, as shown in Fig. 2.

First, a factor graph-based estimation module fuses tactile and proprioceptive feedback to infer the real-time pose of the writing tool ($\mathbf{p}_{obj}$), in-hand contact states ($\mathbf{x}_{int}$, $\mathbf{f}_{int}$), and extrinsic contact conditions ($\mathbf{x}_{ext}$, $\mathbf{f}_{ext}$), without relying on vision. We define a maximum a posteriori (MAP) optimization over these latent variables and solve it incrementally using the Incremental Smoothing and Mapping (iSAM) algorithm from GTSAM [10], enabling real-time and smooth updates as new sensor measurements arrive.

Second, we propose a hybrid arm-hand control framework that combines RL-based in-hand control with compliant wrist adjustments. This framework effectively balances dexterity and stability by decoupling complex finger movements from wrist control, ensuring consistent extrinsic contact. (1) The in-hand policy, trained using PPO, learns six-DoF finger joint displacement commands from tactile observations including fingertip forces/positions, object pose, and extrinsic contact features. The reward function is designed to encourage the agent to manipulate the writing tool effectively while maintaining stable contact, which includes task accuracy, contact stability, and smooth motion. (2) To maintain consistent contact between the pen tip and the writing surface, we design a compliant position-force control mechanism for the wrist. Wrist control operates in three modes: initial position tracking, stable force regulation, and a smooth transition triggered by contact detection, using PD and force feedback controllers to ensure compliant external interactions.

The RL agent is first trained in a simulated Mujoco environment that closely replicates the real robot setup, including tactile sensors, as shown in Fig. 2. To enable effective sim-to-real transfer, it is essential to address critical factors affecting robot performance: (1) High-fidelity tactile signals are simulated in MuJoCo with both spatial and temporal fidelity. Each fingertip tactile sensor is modeled as a distributed array of taxels, and contact is computed with dropout smoothing (zero-hold), signal rounding, and gain scaling. (2) System identification through calibration of tactile force-voltage mappings and joint control dynamics is using optimization (e.g., CMA-ES [11]). (3) Domain randomization with variable object properties and trajectory diversity is applied to improve policy robustness. These ensure consistent deployment in real-world setups.

## III. CONCLUSION

We present a hybrid arm–hand coordination framework that combines reinforcement learning with compliant force–position control to achieve dexterous in-hand writing under extrinsic contact. Fingertip tactile sensors enable real-time estimation of external forces and contact positions without fixed gripper–object contact, allowing stable and precise manipulation of tools with diverse properties. Through careful calibration of joint dynamics and tactile sensing, together with domain randomization, the framework achieves reliable sim-to-real transfer. Extensive experiments in simulation and real-world settings confirm robust performance across varying stiffness, friction, and contact conditions. Detailed results are available at https://inhandwriting.github.io/ , and the work has been published in IEEE Robotics and Automation Letters [12].

Overall, this work lays the foundation for a broad range of robotic manipulation tasks in unstructured and dynamic environments, such as tool usage and assembly, advancing the capabilities of human-like dexterity in robotics.

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
