# OpenReview forum: "Tactile-Driven Dexterous In-Hand Writing via Extrinsic Contact Sensing"
_IEEE.org/IROS/2025/Workshop/Tactile_Sensing — IROS 2025 Workshop Tactile Sensing OralPoster_

### Official Review · Reviewer_84LA · 2025-09-21
**Great work!**

**Rating:** 10
**Confidence:** 5

**Review:**

The paper demonstrates a sim-to-real-based approach to contact-rich tool manipulation with dense touch sensing. The authors validated the system's tool trajectory tracking performance in both simulation and real world. The result is very impressive.

I am a bit curious about the performance on objects with rough edges (e.g. pencils with hexagon section) or more curved surface (screwdriver). It would be great to have them in a more formal paper.

---

### Official Review · Reviewer_XWnU · 2025-09-22
**Interesting problem with compelling real-world demonstration**

**Rating:** 8
**Confidence:** 5

**Review:**

This is solid a system contribution that shows how to utilize a three-finger hand with tactile sensing to perform a contact-rich manipulation task that involves intricate extrinsic contact, that is, in-hand writing. The work is well executed, and the real-world results are impressive, in particular the sim2real performance.

I would like to see more details about the MuJoCo simulation. Are the authors using the default touch sensors in MuJoCo, or did they have to implement their own version? It would be a great contribution to release your code to the community, which has not seen many instances of sim2real transfer with tactile sensing on general-purpose simulators.